# Attention-Only Transformers via Unrolled Subspace Denoising

**Peng Wang**[1]  **Yifu Lu**[1]  **Yaodong Yu**[2]  **Druv Pai**[2]  **Qing Qu**[1]  **Yi Ma**[3]

## Abstract

Despite the popularity of transformers in practice, their architectures are empirically designed and neither mathematically justified nor interpretable. Moreover, as indicated by many empirical studies, some components of transformer architectures may be redundant. To derive a fully interpretable transformer architecture with only necessary components, we contend that the goal of representation learning is to compress a set of noisy initial token representations towards a mixture of low-dimensional subspaces. To compress these noisy token representations, an associated denoising operation naturally takes the form of a multi-head (subspace) self-attention. By unrolling such iterative denoising operations into a deep network, we arrive at a highly compact architecture that consists of *only* self-attention operators with skip connections at each layer. Moreover, we show that each layer performs highly efficient denoising: it improves the signal-to-noise ratio of token representations *at a linear rate* with respect to the number of layers. Despite its simplicity, extensive experiments on vision and language tasks demonstrate that such a transformer achieves performance close to that of standard transformer architectures such as GPT-2 and CRATE.

## 1. Introduction

Over the past decade, transformers (Vaswani et al., 2017) have achieved remarkable empirical success across various modern machine learning applications, including large language models (LLMs) (Brown et al., 2020; Devlin, 2018), vision generative models (Bao et al., 2023; Chen et al., 2020),

and reinforcement learning (Chen et al., 2021). Transformer architectures are generally constructed by stacking multiple identical layers designed to process and learn from data. Each layer is composed of several interacting components arranged in a specific sequence, including multi-head self-attention operators, layer normalization, multi-layer perceptron (MLP) networks, and skip connections. In practice, transformers, such as BERT (Devlin, 2018) and GPT-4 (Achiam et al., 2023), are highly deep, often with dozens or even hundreds of layers, and significantly over-parameterized, containing millions or even billions of parameters. This considerable depth and over-parameterization endow transformers with impressive learning capabilities, allowing them to capture complex patterns and relationships.

Despite the remarkable success of transformers, their deep and over-parameterized architecture renders them "black boxes", hindering an understanding of their inner mechanism. To address this challenge, a common approach involves systematically removing or modifying certain components in transformers to simplify the architecture; see, e.g., Dong et al. (2021); Alcalde et al. (2024); Noci et al. (2024); Geshkovski et al. (2023a); Geva et al. (2020); Guo et al. (2024). For example, Alcalde et al. (2024) studied pure-attention hard-max transformers with skip connections and showed that the output converges to a clustered equilibrium as the number of layers goes to infinity. Noci et al. (2024) analyzed a modified softmax-based attention model with skip connections, demonstrating that the limiting distribution can be described by a stochastic differential equation. These studies indicate that the most basic components of transformers are self-attention layers and skip connections. Although existing studies have provided valuable insights into different components of transformers, few of them elucidate the underlying mechanisms by which transformers process and transform input into output across layers.

Existing empirical studies suggest that some components of transformers may not be essential and can be removed or modified without compromising performance. For example, He & Hofmann (2024) empirically demonstrated that transformer architecture can be simplified by removing components such as skip connections, value matrix, and normalization layers without degrading performance. Additionally, Sukhbaatar et al. (2019) investigated the effects of removing MLP blocks from transformers and augmenting

[1]Department of Electrical Engineering and Computer Science, University of Michigan, Ann Arbor, USA [2]Department of Electrical Engineering and Computer Science, University of California, Berkeley, USA [3]Institute of Data Science & School of Computing and Data Science, University of Hong Kong, Hong Kong, China. Correspondence to: Peng Wang <peng8wang@gmail.com>.

*Proceedings of the $42^{nd}$ International Conference on Machine Learning*, Vancouver, Canada. PMLR 267, 2025. Copyright 2025 by the author(s).

the self-attention layers to play a similar role to MLP blocks, showing that performance can be preserved. Similarly, Pires et al. (2023) examined the potential for reducing the frequency of MLP layers in transformers. Other works also studied other simplifications of transformers, such as linear attentions (Katharopoulos et al., 2020) and shared-QK attentions (Kitaev et al., 2020). Based on these discussions, this work focuses on addressing the following question:

*Can we design a minimalistic transformer architecture consisting of fully interpretable layers that achieves performance close to that of standard transformers?*

### 1.1. Related Works

**Existing studies of self-attention mechanisms.** One main factor of the power of transformers is the self-attention layers, which enable the model to capture long-range dependencies and contextual relationships between tokens by weighing token relationships across the input sequence (Vaswani et al., 2017). To explore the mechanism behind self-attention, numerous studies have investigated the performance of pure self-attention networks, often incorporating only one additional component to prevent rank collapse and maintain expressiveness; see, e.g., Dong et al. (2021); Geshkovski et al. (2023a;b); Wu et al. (2024). We refer the reader to Appendix A for more discussions.

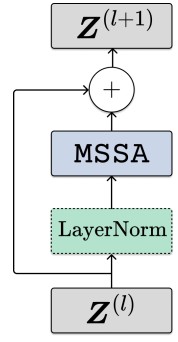

Figure 1. **Each layer of our proposed transformer architecture.**

**Network architecture design via unrolled optimization.** Several lines of work have proposed that the success of modern deep networks largely stems from their ability to transform the raw data into compact and structured representations, which facilitates downstream tasks (Chan et al., 2022; Chen et al., 2023; Ma et al., 2022; Yu et al., 2024; Huh et al., 2024). A principled and interpretable approach to learning such representations is to construct an architecture that incrementally transforms tokens into these representations via unrolling optimization steps as layers of a deep network (Chan et al., 2022; Monga et al., 2021; Wang et al., 2016; Yu et al., 2024; Zhang & Ghanem, 2018). Appendix A contains more discussion on this point.

**Linear representation and superposition hypotheses.** Recent empirical studies of language tasks have raised the "*linear representation hypothesis*", which posits that token representations can be linearly encoded as one-dimensional feature vectors in the activation space of LLMs (Jiang et al., 2024; Park et al., 2024b), and "*superposition hypothesis*", which further hypothesizes that token representations are a sparse linear combination of these feature vectors (Elhage et al., 2022; Yun et al., 2021). Building on these hypotheses, various approaches have been proposed to understand and utilize token representations. For example, Templeton (2024) employed sparse autoencoders to decompose the token representations of Claude 3 Sonnet into more interpretable components. Luo et al. (2024) leveraged sparse dictionary learning to explore token representations, decomposing them into interpretable components based on a concept dictionary. Recently, Engels et al. (2025) conjectured that token representations in LLMs are the sum of many sparse multi-dimensional features. This conjecture is supported by their experiments on GPT-2 and Mistral 7B, where they used sparse autoencoders to identify multi-dimensional features. Notably, all of these empirical studies conclude that *the token representations lie on a union of (possibly many) low-dimensional subspaces*.

### 1.2. Our Contributions

Based on the above discussions and motivated by referenced empirical findings, we propose a simple yet evocative model for the structure of token representations in trained transformers. Specifically, we model the underlying distribution of token representations as a mixture of low-rank Gaussians, each supported on a subspace and corrupted by noise (see Definition 2.1). Then, the goal of representation learning is to denoise a set of noisy initial token representations towards the corresponding subspaces. Our contributions are summarized as follows:

- **Attention-only transformer via unrolled optimization.** Under the mixture of low-rank Gaussian model, we interpret multi-head (subspace) self-attention as a denoising operator, which compresses noisy token representations into their corresponding supporting subspaces. By iteratively unrolling the multi-head (subspace) self-attention operator, we construct a new transformer architecture with a streamlined design, consisting of only self-attention layers with skip connections (see Figure 1).[1] This design is much simpler compared to standard transformers.

- **Theoretical guarantees for the proposed transformer.** To quantify the denoising performance of the proposed transformer, we introduce a signal-to-noise (SNR) metric (see Eq. (8)) for the token representations. We prove that each layer of the proposed transformer improves the SNR at a linear rate when the initial token representations are sampled from a noisy mixture of low-rank Gaussians (see Theorem 3.1). This indicates that the multi-head (subspace) self-attention operator is highly effective in denoising token representations towards their corresponding subspaces.

---

[1]In practice, LayerNorm layers may be added to enhance performance.

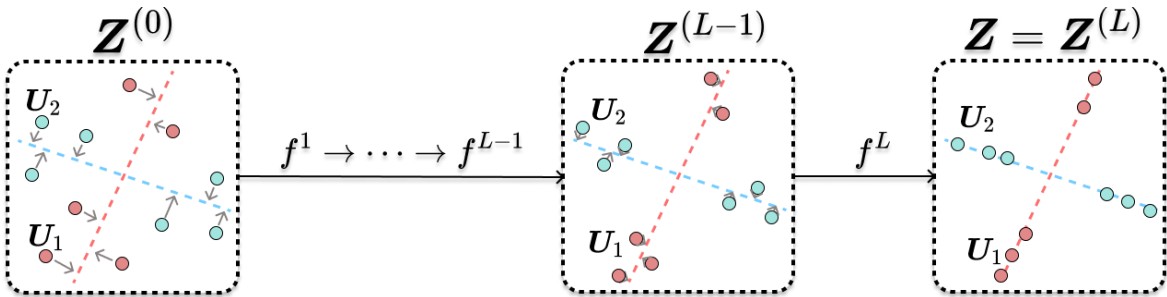

*Figure 2.* **Layers of a transformer gradually denoise token representations $\boldsymbol{Z}^{(l)}$ towards their corresponding subspaces.**

- **Understanding the roles of self-attention and MLP layers.** Notably, the proposed transformer is a valuable model for understanding the mechanism of attention, as it ablates the effect of MLP layers. Moreover, comparing the proposed transformer to standard transformers provides insights into the specific role and empirical benefits of the MLP layers in different tasks, such as for images and texts (see experiments in Section 4).

Finally, we conduct extensive experiments on both vision and language tasks, including supervised image classification, causal language modeling, and in-context learning, to complement our theory and demonstrate the potential of our proposed transformer architecture. *We emphasize that the goal of our experiments is not to strive for state-of-the-art performance for these tasks. Instead, they are intended to validate our theory about the components of the transformer.*

**Notation.** Given an integer $n$, we denote by $[n]$ the set $\{1, \ldots, n\}$. Given a vector $\boldsymbol{a}$, let $\|\boldsymbol{a}\|$ denote the Euclidean norm of $\boldsymbol{a}$ and $\mathrm{diag}(\boldsymbol{a})$ denote the diagonal matrix with $\boldsymbol{a}$ as its diagonal. Given a matrix $\boldsymbol{A}$, let $\|\boldsymbol{A}\|$ denote the spectral norm of $\boldsymbol{A}$, $\|\boldsymbol{A}\|_F$ denote the Frobenius norm, and $a_{ij}$ denote the $(i, j)$-th element. For sequences of positive numbers $\{a_n\}$ and $\{b_n\}$, we write $a_n \lesssim b_n$ or $b_n \gtrsim a_n$ if there exists an absolute constant $C > 0$ such that $a_n \leq C b_n$. Given a constant $\tau > 0$, we define $\mathbb{I}(x > \tau) = 1$ if $x > \tau$ and $\mathbb{I}(x > \tau) = 0$ otherwise. We use $\mathcal{O}^{n \times d}$ to denote the set of all $n \times d$ matrices that have orthonormal columns.

## 2. Technical Approach and Justification

In this section, we introduce the basic setup of transformers for learning representations from real-world data. Real-world data, such as images, videos, and text, are often modeled as random samples drawn from a high-dimensional probability distribution with low-dimensional intrinsic structures (Wright & Ma, 2022). Instead of directly inputting raw data samples into transformers, a common preprocessing step is to convert each sample into a sequence of tokens, where each token represents a localized segment of the data, such as an image patch, a snippet of text, or a frame in a video. Consequently, the input to

transformers is typically a sequence of tokens denoted as $\boldsymbol{X} = [\boldsymbol{x}_1, \ldots, \boldsymbol{x}_N] \in \mathbb{R}^{D \times N}$. Then, the goal of transformers is to learn a map $f$ that transforms these tokens into structured and compact token representations that facilitate downstream tasks, such as classification (Dosovitskiy et al., 2021) and generation (Saharia et al., 2022), by capturing the underlying patterns in the data.

### 2.1. Unrolled Optimization for Token Representations

In this subsection, we introduce how to learn token representations using the approach of unrolling optimization algorithms (Chan et al., 2022; Gregor & LeCun, 2010; Monga et al., 2021; Sun et al., 2019; Wang et al., 2016; Yu et al., 2024; Zhang & Ghanem, 2018). Specifically, this approach constructs each layer of a neural network according to a step of an iterative optimization algorithm. That is, the network's architecture is designed to implement a specific optimization algorithm, where each layer corresponds to a single iterative step. By unrolling the algorithm, we construct a "white-box" transformer architecture as a multi-layer neural network that incrementally transforms input tokens into structured and compact representations. This iterative process can be described as follows:

$$f : \boldsymbol{X} \xrightarrow{f^0} \boldsymbol{Z}^{(0)} \xrightarrow{f^1} \cdots \xrightarrow{f^l} \boldsymbol{Z}^{(l)} \xrightarrow{f^{l+1}} \cdots \xrightarrow{f^L} \boldsymbol{Z}^{(L)} =: \boldsymbol{Z},$$

where $f^0 : \mathbb{R}^{D \times N} \to \mathbb{R}^{d \times N}$ is a pre-processing mapping (e.g., positional encoding, token embedding) that transforms input tokens $\boldsymbol{X} \in \mathbb{R}^{D \times N}$ to initial token representations $\boldsymbol{Z}^{(0)} \in \mathbb{R}^{d \times N}$, $f^l : \mathbb{R}^{d \times N} \to \mathbb{R}^{d \times N}$ denotes an iterative step or *layer*, and $\boldsymbol{Z}^{(l)}$ denotes the token representations at the $l$-th layer for each $l \in [L]$. Then, a key question is *how to design the operator $f^l$ at each layer to learn meaningful token representations in a principled manner*.

### 2.2. A Model for Token Representations

Before we design such an operator $f^l$, we model the structure of token representations in pretrained LLMs. Notably, extensive works (Templeton, 2024; Luo et al., 2024; Engels et al., 2025) have empirically demonstrated that token representations in trained LLMs usually approximately lie in a union of low-dimensional subspaces. These subspaces

encode distinct semantic meanings, capturing various linguistic or contextual features that contribute to the model's overall understanding and performance. This motivates us to model the token representations as follows:

**Definition 2.1.** Let $C_1, \ldots, C_K$ be a partition of the index set $[N]$ and $\boldsymbol{U}_k \in \mathcal{O}^{d \times p_k}$ denote the orthonormal basis of the $k$-th subspace for each $k \in [K]$. We say that the token representations $\{\boldsymbol{z}_i\}_{i=1}^N \subseteq \mathbb{R}^d$ are sampled from a mixture of noisy low-rank Gaussian distributions if for each $k \in [K]$,

$$\boldsymbol{z}_i = \underbrace{\boldsymbol{U}_k \boldsymbol{a}_i}_{\text{signal}} + \underbrace{\sum_{j \neq k}^K \boldsymbol{U}_j \boldsymbol{e}_{i,j}}_{\text{noise}}, \ \forall i \in C_k, \qquad (1)$$

where $\boldsymbol{a}_i \overset{i.i.d.}{\sim} \mathcal{N}(\boldsymbol{0}, \boldsymbol{I}_{p_k})$ and $\boldsymbol{e}_{i,j} \overset{i.i.d.}{\sim} \mathcal{N}(\boldsymbol{0}, \delta^2 \boldsymbol{I}_{p_j})$ for all $i \in C_k$ and $k \in [K]$, $\{\boldsymbol{a}_i\}$ and $\{\boldsymbol{e}_{i,j}\}$ are respectively mutually independent, and $\{\boldsymbol{a}_i\}$ is independent of $\{\boldsymbol{e}_{i,j}\}$.

Before we proceed, let us make some remarks on this model.

- **An idealized model for token representations.** This model serves as an idealized framework for approximating token representations in real-world pretrained LLMs. It assumes that the token representations are sampled from a mixture of multiple low-rank Gaussian distributions with noise. Under this model, the goal of representation learning is to compress a set of noisy initial token presentations into the corresponding subspace. We should point out that in real-world applications, where token representations exhibit more complicated structures, the goal of representation learning is to find a compact and structured representation via compressing token sets, as argued in Yu et al. (2024). In addition, this model has been widely used in other machine learning problems, such as subspace clustering (Wang et al., 2022; Elhamifar & Vidal, 2013) and diffusion models (Wang et al., 2024).

- **Connections to hypotheses on token representations.** This model aligns well with two well-established hypotheses about the structure of token representations in pretrained LLMS: the "linear representation hypothesis" (Jiang et al., 2024; Park et al., 2024b) and the "superposition hypothesis" (Elhage et al., 2022; Yun et al., 2021; Arora et al., 2018). The linear representation hypothesis posits that token representations in LLMs lie in low-dimensional linear subspaces that encode semantic features. Similarly, the superposition hypothesis suggests that these representations can be approximately expressed as a sparse linear combination of these feature vectors. In the context of our model, each basis $\boldsymbol{U}_k$ of the subspaces can be interpreted as a set of semantic features, where each feature corresponds to a specific aspect of the token's meaning. Token representations are then approximately

expressed as sparse linear combinations of these subspace bases, capturing the essential semantic components of the token while ignoring irrelevant dimensions.

### 2.3. Denoising Operator for Token Representations

Now, we introduce a denoising operator to compress token representations into the corresponding subspace when the initial token representations $\boldsymbol{Z}^{(0)}$ is generated according to Definition 2.1. To simplify our development, we assume that the subspaces in Definition 2.1 are orthogonal to each other, i.e., $\boldsymbol{U}_k^T \boldsymbol{U}_j = \boldsymbol{0}$ for all $k \neq j$. Note this assumption is not restrictive, as in high-dimensional spaces, random low-dimensional subspaces are incoherent to each other with high probability, i.e., $\boldsymbol{U}_k^T \boldsymbol{U}_j \approx \boldsymbol{0}$ (Wright & Ma, 2022).[2]

**Multi-head subspace self-attention.** Without loss of generality, we rearrange the initial token representations $\boldsymbol{Z}^{(0)}$ such that those from the same subspace are concatenated together, i.e., $\boldsymbol{Z}^{(0)} = [\boldsymbol{Z}_1^{(0)}, \ldots, \boldsymbol{Z}_K^{(0)}]$ with

$$\boldsymbol{Z}_k^{(0)} = \boldsymbol{U}_k \boldsymbol{A}_k + \sum_{j \neq k} \boldsymbol{U}_j \boldsymbol{E}_{k,j}, \ \forall k \in [K].$$

Here, the columns of $\boldsymbol{Z}_k^{(0)}$ denote the token representations from the $k$-th subspace for each $k \in [K]$, the columns of $\boldsymbol{A}_k \in \mathbb{R}^{p_k \times N_k}$ consists of $\{\boldsymbol{a}_i\}_{i \in C_k}$, and the columns of $\boldsymbol{E}_{k,j} \in \mathbb{R}^{p_j \times N_k}$ consists of $\{\boldsymbol{e}_{i,j}\}_{i \in C_k}$ for each $k \in [K]$ with $N_k = |C_k|$ for each $k \in [K]$. Obviously, projecting token representations onto their corresponding subspace helps to separate the signal from the noise components using $\boldsymbol{U}_k^T \boldsymbol{U}_j = \boldsymbol{0}$ for all $k \neq j$, i.e.,

$$\boldsymbol{U}_k \boldsymbol{U}_k^T \boldsymbol{Z}_\ell^{(0)} = \begin{cases} \boldsymbol{U}_k \boldsymbol{A}_k, & \text{if } \ell = k, \\ \boldsymbol{U}_k \boldsymbol{E}_{\ell,k}, & \text{if } \ell \neq k. \end{cases} \qquad (2)$$

To denoise the token representations from $k$-th subspace, we compute the similarity of projected token representations via $(\boldsymbol{U}_k^T \boldsymbol{Z})^T (\boldsymbol{U}_k^T \boldsymbol{Z})$ and verify that the similarity between projected token representations from the $k$-th subspace is high, while the similarity between other pairs of projected token representations is low when $\delta < 1$. Then, we convert it to a distribution of membership with function $\varphi$, such as hard-thresholding or soft-max functions, and denoise the token representations towards to the corresponding subspace using this membership. Now, we formalize the considered operator as follows: for each $l = 0, 1, \ldots, L-1$,

$$\boldsymbol{Z}^{(l+1)} = \boldsymbol{Z}^{(l)} + \eta \, \text{MSSA}(\boldsymbol{Z}^{(l)}), \qquad (3)$$

where $\eta > 0$ is the denoising step size, $\varphi(\cdot) : \mathbb{R}^{d \times N} \to \mathbb{R}^{d \times N}$ is an operator applied column-wise, and

$$\text{MSSA}(\boldsymbol{Z}) = \sum_{k=1}^K \boldsymbol{U}_k \boldsymbol{U}_k^T \boldsymbol{Z} \varphi\left(\boldsymbol{Z}^T \boldsymbol{U}_k \boldsymbol{U}_k^T \boldsymbol{Z}\right). \qquad (4)$$

---

[2]One may straightforwardly generalize our results to non-orthogonal subspaces, with slightly more sophisticated analysis.

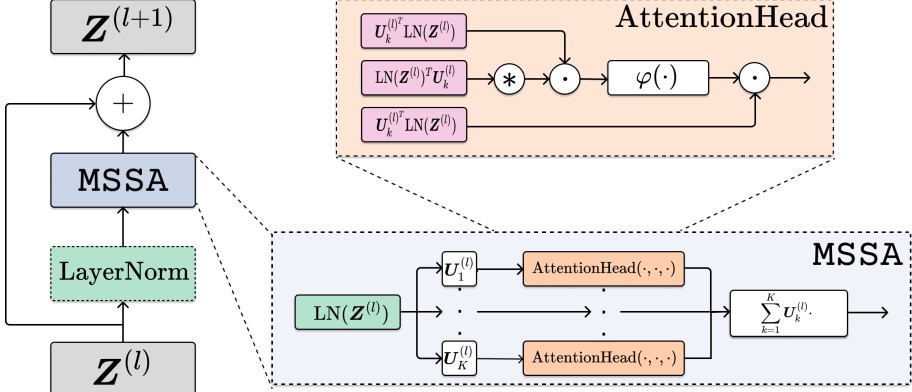

*Figure 3.* **The attention-only transformer (AoT) architecture.** Each layer consists of the MSSA operator and a skip connection. Additionally, LayerNorm can be incorporated to enhance performance. In practice, backpropagation is applied to train the model parameters.

Notably, the operator in (4), referred to as the *multi-head subspace self-attention* (**MSSA**), is first proposed by (Yu et al., 2024; 2023) to approximately optimize the compression term of the sparse rate reduction objective for constructing a transformer-like architecture. It is worth noting that Yu et al. (2023); Pai et al. (2023) showed that the negative compression gradient of the objective points from the token representation to the corresponding subspace. However, they did not study the quantitative denoising efficiency of the MSSA operator (4).

**Connections to multi-head self-attention.** Notably, the denoising operator (4) is essentially a special instance of multi-head self-attention (**MHSA**) implemented with a skip connection in transformers. Specially, the multi-head self-attention is of the following form:

$$\text{MHSA}(\boldsymbol{Z}) = \boldsymbol{W}^O \begin{bmatrix} \text{head}_1 \\ \vdots \\ \text{head}_K \end{bmatrix} \quad (5)$$

where $\boldsymbol{W}_k^Q, \boldsymbol{W}_k^K, \boldsymbol{W}_k^V$ are learnable weight matrices for queries, keys, and values for head $k$, $\boldsymbol{W}^O$ is another learnable weight matrix, and

$$\text{head}_k = (\boldsymbol{W}_k^V)^T \boldsymbol{Z} \varphi(\boldsymbol{Z}^T \boldsymbol{W}_k^Q (\boldsymbol{W}_k^K)^T \boldsymbol{Z}).$$

Comparing the MHSA operator with the MSSA operator in (4), by setting $\boldsymbol{W}_k^Q = \boldsymbol{W}_k^K = \boldsymbol{W}_k^V = \boldsymbol{U}_k$ and $\boldsymbol{W}^O = [\boldsymbol{U}_1, \dots, \boldsymbol{U}_K]$ in (5), we can obtain the MSSA operator in (4). In this special case, MHSA can be interpreted as a denoising operation onto different subspaces. However, token representations in state-of-the-art large models are inherently more complex than the simplified structures assumed in Definition 2.1. In practice, these token representations are subject to a variety of factors such as noise, context dependence, and intricate dependencies that make their structure more dynamic and multifaceted. In this context, using the more flexible MHSA mechanism may provide a better way to denoise these complex token representations. To sum up, while state-of-the-art models necessitate the use of more advanced mechanisms like MHSA to effectively denoise and optimize token representations, the model in Definition 2.1 offers a useful framework for understanding token representations in an idealized yet evocative setting.

## 3. Main Results

In this section, we formally present an attention-only transformer architecture using unrolled optimization and provide a theoretical guarantee on its denoising performance.

### 3.1. Attention-Only Transformer

Armed with the setup in Section 2, we formally introduce the proposed attention-only transformer architecture. Specifically, by unrolling the iterative optimization steps (3) as layers of a deep network, we construct a transformer architecture in Figure 3. Each layer of the proposed architecture consists only of the MSSA operator and a skip connection. To enhance the model's performance, we may additionally incorporate LayerNorm before the MSSA operator to improve performance in practice. The complete architecture is built by stacking such layers, along with essential task-specific pre-processing and post-processing steps, such as positional encoding, token embedding, and a final task-specific head to adapt to different applications. Notably, if we apply the same procedure to (5), we obtain an attention-only transformer that only consists of the MHSA operator.

**Comparison to standard transformers.** Generally speaking, the standard decoder-only transformer architecture is composed of the following key components (Brown et al., 2020; Radford et al., 2019): (1) positional encoding, (2) multi-head QKV self-attention mechanisms, (3) feed-forward MLP networks, (4) layer normalization, and (5) residual connections. In contrast, our proposed transformer

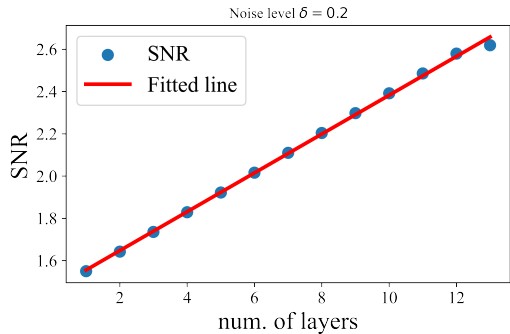 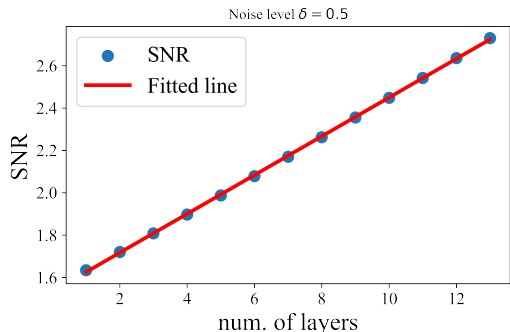

*Figure 4.* **Denoising performance of the attention-only transformer.** Here, we sample initial token representations from a mixture of low-rank Gaussians in Definition 2.1. Then, we apply (4) to update token representations and report the SNR at each layer. Left: noise level $\delta = 0.2$. Right: noise level $\delta = 0.5$.

architecture adopts a streamlined design by incorporating several key simplications. Specifically, it employs shared-QKV subspace self-attention mechanisms, excludes MLP layers, and reduces the frequency of LayerNorm.

**Differences from previous works on attention-only transformers.** In the literature, some theoretical works have studied attention-only transformers. For example, Dong et al. (2021); Wu et al. (2024) showed that pure-attention transformers with skip connections or LayerNorm can prevent rank collapse. Additionally, Alcalde et al. (2024) studied the clustering behavior of attention-only hardmax transformers. While these studies contribute significantly to our understanding of the role of self-attention in transformers, they lack empirical validation and practical implications. In contrast to these works, we not only show that each layer of the proposed attention-only transformer can denoise token representations but also conduct experiments on real-world language and vision tasks to demonstrate the potential.

**The role of backward propagation.** Notably, our approach constructs a transformer architecture in the forward pass by interpreting each layer as a denoising operator, conditioned on the assumption that the subspace bases $\{U_k\}_{k=1}^K$ are known. However, in practice, these subspace bases are unknown and need to be learned gradually via backpropagation. Hence, the forward denoising operator (4) at the $l$-th layer becomes as follows: For each $l = 0, 1, \ldots, L-1$,

$$
\begin{aligned}
Z^{(l+1)} = Z^{(l)} + \eta \sum_{k=1}^K U_k^{(l)} U_k^{(l)^T} Z^{(l)} \\
\varphi\left(Z^{(l)^T} U_k^{(l)} U_k^{(l)^T} Z^{(l)}\right).
\end{aligned}
$$

Now, the parameters $\{U_k^{(l)}\}$ depend on the layer index $l$ and may be different across layers. These matrices are learned through end-to-end training via backpropagation.

### 3.2. Denoising via Attention-Only Transformer

In this subsection, we study the denoising performance of the proposed transformer when the initial token representations are sampled from a mixture of low-rank Gaussians as introduced in Definition 2.1. To quantify the denoising performance, we define the signal-to-noise ratio (SNR) for each cluster of the token representations at the $l$-th layer as

$$
\text{SNR}(Z_k^{(l)}) := \frac{\|U_k U_k^T Z_k^{(l)}\|_F}{\|(I - U_k U_k^T) Z_k^{(l)}\|_F}, \ \forall k \in [K]. \quad (6)
$$

To simplify our analysis, we assume that $p = p_1 = \cdots = p_K$, $N_1 = \cdots = N_K = N/K$, and

$$
\begin{bmatrix} U_1 & \cdots & U_K \end{bmatrix} \in \mathcal{O}^{d \times Kp}. \quad (7)
$$

With the above setup, we now prove the following theorem.

**Theorem 3.1.** *Let $Z^{(0)}$ be generated according to Definition 2.1 and $Z^{(l)}$ be generated according to (3) for each $l \in [L]$. Here, $\varphi(x) = h(\sigma(x))$, $\sigma : \mathbb{R}^N \to \mathbb{R}^N$ is the soft-max function, and $h : \mathbb{R}^N \to \mathbb{R}^N$ is an element-wise thresholding function with $h(x) = \tau \mathbb{I}\{x > \tau\}$ for each $i \in [N]$. Suppose that $p \gtrsim \log N$, $\delta \lesssim \sqrt{\log N}/\sqrt{p}$, and*

$$
\tau \in \left(\frac{1}{2}, \frac{1}{1 + N \exp(-9p/32)}\right].
$$

*For sufficiently large $N$, it holds with probability at least $1 - KLN^{-\Omega(1)}$ that for each $l \in [L-1]$,*

$$
\text{SNR}\left(Z_k^{(l+1)}\right) = (1 + \eta\tau)\text{SNR}\left(Z_k^{(l)}\right), \ \forall k \in [K]. \quad (8)
$$

The proof is deferred to Appendix B. Here we comment on the significance of this theorem:

- **Linear denoising performance of the attention-only transformer.** When the initial token representations are sampled from a mixture of low-rank Gaussian distributions with a noise level $O(\sqrt{\log N}/\sqrt{p})$, we show that

each layer of the proposed transformer denoises token representations at a linear rate. This indicates the MSSA operator's efficiency in reducing noise across layers. Notably, our theoretical results are well-supported by experimental observations in Figure 4, which further validate the practical denoising capability of the proposed transformer.

- **Difficulties in analyzing the dynamics of the update (3).** Note that the update (3) is highly nonlinear and complicated. These characteristics lead to intricate interactions among consecutive updates that complicate the analysis of the learning dynamics. Compared to the existing works (Ahn et al., 2023; Zhang et al., 2024; Schlag et al., 2021) that mainly focus on linear self-attention with $\varphi(\cdot)$ being the identify function, our analysis provides more pertinent results for understanding the denoising performance and learning dynamics of attention mechanisms, capturing the *nonlinear* interactions and transformations across the layers of modern transformer architectures.

## 4. Experimental Verification

In this section, we evaluate our proposed *attention-only transformer* (AoT) architecture using the MSSA (denoted by **AoT-MSSA**) and MHSA (denoted by **AoT-MHSA**) operators on both vision and language tasks. Since the model configurations on vision and language tasks are different, we use **AoT-MSSA-V** and **AoT-MHSA-V** to denote the models applied to vision tasks, and **AoT-MSSA-L** and **AoT-MHSA-L** for those applied to language tasks. Due to limited computing resources, the goal of our experiments is not to outperform state-of-the-art transformers but to verify that AoT can achieve comparable performance on both language and vision tasks. In our implementations, we set the operator $\varphi(\cdot)$ in Eq. (4) to be the softmax function.

*Table 1.* Top-1 accuracy on ImageNet: Evaluation of AoT-MSSA-V and comparison to CRATE.

| Models | Accuracy | # of Parameters |
|---|---|---|
| AoT-MSSA-V | 71.7% | 22M |
| CRATE | 79.5% | 39M |

### 4.1. Vision Transformers for Image Classification

In this subsection, we evaluate the performance of AoT as a backbone architecture for supervised image classification on ImageNet and compare it against several state-of-the-art models. To construct the AoT-based model, we adopt the same preprocessing pipeline and classification head as defined in (Yu et al., 2024, Section 4.1.1).

**Comparison between MSSA and CRATE.**   We consider the AoT-MSSA-V model and compare it against the CRATE model in Yu et al. (2024). We employ Lion optimizer

*Table 2.* Top-1 accuracy on ImageNet: Evaluation of AoT-MHSA-V and comparison to ViT.

| Models | Accuracy | # of Parameters |
|---|---|---|
| AoT-MHSA-V | 69.5% | 15M |
| ViT | 72.4 % | 22M |

(Chen et al., 2024) to pre-train the AoT-MSSA-V transformer on ImageNet-21K for 90 epochs and to fine-tune it on ImageNet-1K (Deng et al., 2009) for 50 epochs by minimizing the cross-entropy (CE) loss. We use different hyperparameters for pre-training and fine-tuning. During pre-training, we use a learning rate of $2 \times 10^{-4}$, weight decay of $0.7$, label smoothing with a parameter of $0.2$, and a batch size of $4096$. For fine-tuning, the corresponding values are $5 \times 10^{-4}$, $0.3$, $0.1$, and $2048$, respectively. Standard data augmentation techniques, including random cropping, random horizontal flipping, and random augmentation, are used in our implementation, following the same setup as in Yu et al. (2023).

**Comparison between MHSA and ViT.**   We next train AoT-MHSA-V from scratch on ImageNet-1K for 150 epochs and compare its performance with ViT (Dosovitskiy et al., 2021). The training setup follows the same configuration as above: we use the Lion optimizer with a learning rate of $5 \times 10^{-4}$, a weight decay of $0.1$, label smoothing with a smoothing parameter of $0.1$, a batch size of $2048$, and identical data augmentation strategies.

Based on the above experimental setup, we report the top-1 accuracy of AoT-MSSA-V and CRATE in Table 1, and that of AoT-MHSA-V and ViT in Table 2. Due to the absence of MLP layers in AoT, AoT-based models achieve slightly worse performance comparable to CRATE and ViT while using only nearly half the number of parameters. This result shows the effectiveness of the attention-only architecture. We provide visualization for the self-attention heatmaps of AoT-MSSA-V trained on ImageNet-1K in Figure 7 in Appendix C.3. We observe that each head captures similar semantic meanings across different images, demonstrating the interpretability of our proposed architecture in practice.

### 4.2. Decoder-Only Transformers for Language Tasks

To study the performance of our architecture on language tasks, we consider the widely used Generative Pre-Training (GPT) task (Radford et al., 2019). In the context of causal language modeling, the goal is to predit the next token in a sequence based on the preceding context. To adapt to this task, we modify the AoT architecture by changing the MSSA (resp., MHSA) operator to a causally masked MSSA (resp., MHSA) operator. We follow the same preprocessing and post-processing steps in (Yu et al., 2024,

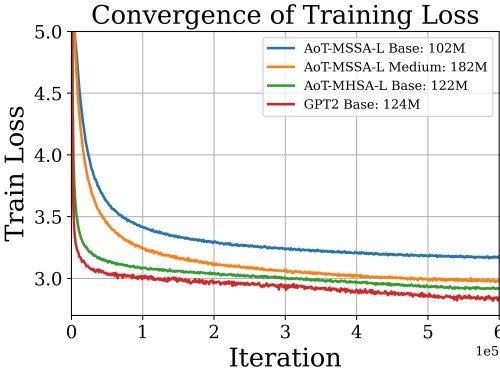 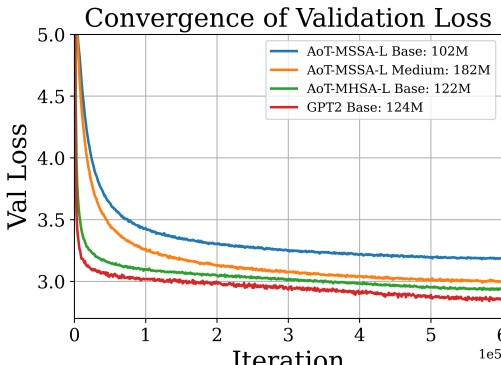

*Figure 5.* **Evaluating models on language tasks.** We plot the training loss (left) and validation loss (right) of the AoT and GPT-2 models pretrained on OpenWebText.

*Table 3.* Zero-shot results on several language benchmark datasets and tasks: Evaluation of different sizes of AoT with the MSSA and MHSA operators and comparison to the GPT2 model.

| Models
# of parameters | LAMBADA
(val loss) ↓ | PTB
(val loss) ↓ | WikiText
(val loss) ↓ | LAMBADA
(acc) ↑ | CBT CN
(acc) ↑ | CBT NE
(acc) ↑ |
|---|---|---|---|---|---|---|
| AoT-MSSA-L Base (102M) | 4.70 | 6.03 | 4.65 | 0.25 | 0.80 | 0.74 |
| AoT-MSSA-L Medium (182M) | 4.47 | 5.08 | 4.22 | 0.29 | 0.84 | 0.77 |
| AoT-MHSA-L Base (122M) | 4.42 | 5.52 | 4.19 | 0.38 | 0.86 | 0.82 |
| GPT-2 Base (124M) | 4.32 | 5.75 | 4.13 | 0.40 | 0.87 | 0.84 |

Section 4.1.4). Our implementation of the GPT-2 type transformers and training pipeline is based on the framework outlined in (Karpathy, 2022).[3]

### 4.2.1. LANGUAGE MODELING

**Pre-training language models.** We pre-train the AoT-MSSA-L and AoT-MHSA-L models of different sizes, along with GPT-2 (see Table 3 for model sizes), on OpenWeb-Text (Gokaslan & Cohen, 2019). We defer the details of the model architectures to Table 4. Here, we train these models over a 1024-token context using the AdamW optimizer (Loshchilov & Hutter, 2019). We plot the training loss and validation loss against the number of training iterations in Figure 5(a) and (b), respectively. We observe that medium- and large-sized AoT-based models achieve training and validation losses comparable to those of the GPT-2 base model. In addition, compared to the GPT-2 base model, the AoT-MHSA-L model is identical to the GPT-2 base model, except for the absence of MLP layers in the architecture. As shown in Figure 5, incorporating MLP layers can accelerate the training process.

**Zero-shot evaluation.** Using the above pre-trained models, we compute the cross-entropy validation loss without training on datasets WikiText (Merity et al., 2017)[4], LAM-

BADA (Paperno et al., 2016)[5], and PTB (Marcus et al., 1993) in Table 3. In addition, we report zero-shot accuracy in Table 3 on LAMBADA for predicting the final word of sentences, as well as on the Children's Book Test (CBT) (Hill et al., 2015), where the task is to choose either common nouns (CN) or named entities (NE) from 10 possible options for an omitted word in a paragraph. We observe that the AoT models with medium and large parameter sizes can achieve comparable performance to the GPT-2 base model. Moreover, we found that adding MLP layers to AoT does not improve the zero-shot performance. These results highlight the potential of attention-only models to achieve competitive results while maintaining interpretability.

### 4.2.2. IN-CONTEXT LEARNING

In-context learning (ICL) refers to the ability of modern language models to perform tasks by using examples provided in the input prompt, along with a new query input, generating outputs without updating the parameters (Brown et al., 2020; Garg et al., 2022; Park et al., 2024a). We evaluate the ICL capabilities of our AoT models and compare their performance with that of GPT-2 (Radford et al., 2019). Each model is trained from scratch on specific tasks, including linear and sparse linear regressions. We mainly follow the setup in (Garg et al., 2022) to train models to

---

[3]https://github.com/karpathy/nanoGPT.git

[4]For WikiText2 and WikiText103 (Merity et al., 2017), the test splits are the same, so we merge them as a single dataset referred

to as WikiText.

[5]To obtain the accuracy on LAMBADA dataset, we use greedy decoding.

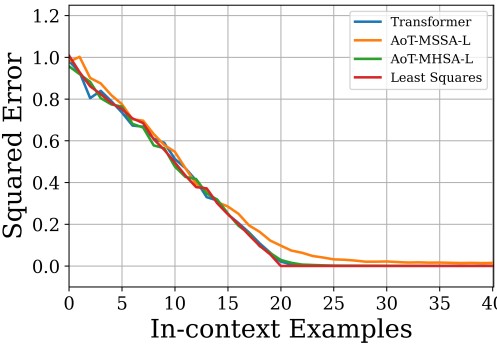 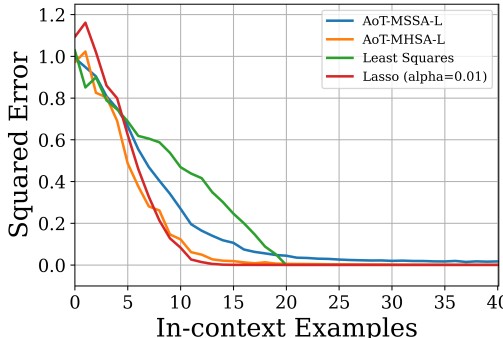

*Figure 6.* **Evaluating models on in-context learning tasks.** We plot the normalized squared error as a function of the number of in-context examples for linear regression (left) and sparse linear regression (right) tasks.

learn linear functions in context. Specifically, for a specific function class $\mathcal{G}$, we generate random prompts by sampling a function $g \in \mathcal{G}$ from distribution $\mathcal{D}_{\mathcal{G}}$ over functions random inputs $\boldsymbol{x}_1, \ldots, \boldsymbol{x}_N \in \mathbb{R}^d$ i.i.d. from $\mathcal{D}_{\mathcal{X}}$ over inputs. To evaluate the inputs on $g$, we create a prompt $P = (\boldsymbol{x}_1, g(\boldsymbol{x}_1), \ldots, \boldsymbol{x}_N, g(\boldsymbol{x}_N))$. We train the model $f_{\boldsymbol{\theta}}(\cdot)$ to minimize the expected loss over all prompts prefixes:

$$\min_{\boldsymbol{\theta}} \mathbb{E}_P \left[ \frac{1}{N} \sum_{i=1}^{N-1} \left( f_{\boldsymbol{\theta}}(P^i) - g(\boldsymbol{x}_i) \right)^2 \right], \qquad (9)$$

where $P^i$ is the prompt prefix up to the input $i$-th in-context example $P = (\boldsymbol{x}_1, g(\boldsymbol{x}_1), \ldots, \boldsymbol{x}_i)$.

**Tasks.** We consider both linear functions and sparse linear functions with dimension $d = 20$. The in-context examples $\boldsymbol{x}_i$ are sampled from the isotropic Gaussian distribution. For linear functions, we define $\mathcal{G} = \{g : g(\boldsymbol{x}) = \boldsymbol{w}^T \boldsymbol{x}\}$, where $\boldsymbol{x}$ is sampled from the isotropic Gaussian distribution as well. For sparse linear functions, the setup is similar, but with a modification: only 3 coordinates in the vector $\boldsymbol{w}$ are set as non-zero, while the remaining ones are set to be zero.

**Training and evaluation.** For all experiments, we set the number of heads to 8 and the embedding size to 128. We present the model configurations in Table 5 in Appendix C. To train the model, we sample a batch of random prompts with size 64 and train the models for 50,000 iterations using Adam optimizer (Kingma & Ba, 2014). We evaluate models using same $\mathcal{D}_{\mathcal{G}}$ and $\mathcal{D}_{\mathcal{X}}$ to sample 1280 prompts. We refer the reader to (Park et al., 2024a) for more details. We plot the estimation error against the in-context samples in Figure 6. We observe that our AoT architecture can in-context learn linear functions and sparse linear functions, achieving performance close to that of the GPT-2 transformer.

## 5. Conclusion

In this work, we proposed a new and minimalistic transformer architecture by interpreting each layer as a subspace denoising operator to token representations, where these representations are assumed to be sampled from a mixture of low-rank Gaussians. Remarkably, this simple architecture consists of multi-head (subspace) self-attention and skip connections at each layer, without MLP layers at all. We have rigorously proven that each such layer improves the signal-to-noise ratio of token representations at a linear rate with respect to the number of layers. Extensive experiments on both language and vision tasks demonstrate that this simplified architecture achieves performance comparable to that of standard transformers. Our theoretical and empirical findings suggest that subspace denoising via attention heads is the core mechanism underlying transformer effectiveness, with MLP layers contributing only marginal performance gains. We believe this work lays a foundation for future exploration of more efficient and principled architectural designs.

## Impact Statement

This paper presents work whose goal is to advance the field of machine learning. Both theoretical analysis and experimental evaluation presented in this paper aim to help people better understand the attention layer in a standard transformer architecture. Except for values to academics, we do not anticipate any social or ethical implications.

## Acknowledgment

Peng Wang and Qing Qu would like to acknowledge support from the NSF grant #2402950. Druv Pai would like to acknowledge support from the UC Berkeley College of Engineering Fellowship. Yi Ma would like to acknowledge support from the joint Simons Foundation-NSF DMS grant #2031899, the ONR grant N00014-22-1-2102, the NSF grant #2402951, and also support from and the HKU startup, the Hong Kong Center for Construction Robotics Limited (HKCRC) Award 052245, and JC Club of Hong Kong.

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

To simplify our development, we introduce some further notations. We use $\mathrm{BlkDiag}(\boldsymbol{X}_1, \ldots, \boldsymbol{X}_K)$ to denote a block diagonal matrix whose diagonal blocks are $\boldsymbol{X}_1, \ldots, \boldsymbol{X}_K$.

## A. Related Literature

**Existing studies on self-attention mechanisms.** It is widely believed that the power of transformers primarily stems from their self-attention layers, which enable the model to capture long-range dependencies and contextual relationships between tokens by dynamically weighing token relationships across the input sequence (Tsai et al., 2019; Vaswani et al., 2017). To explore the mechanism behind self-attention, numerous studies have investigated the performance of pure self-attention networks, often incorporating only one additional component to prevent rank collapse and maintain expressiveness. For example, Dong et al. (2021) showed that in pure-attention transformers without skip connections and MLP layers, token representations collapse exponentially to a rank-1 matrix across layers. They also showed that self-attention networks with skip connections prevent rank collapse. Geshkovski et al. (2023a;b) have studied the dynamics of multi-head self-attentions and characterized clustering behaviors of learned representations. Recently, Wu et al. (2024) showed that pure self-attention networks with LayerNorm can prevent rank collapse. While these studies have advanced the theoretical understanding of self-attention mechanisms in simplified transformer architectures, they cannot provide any empirical validation on real-world vision or language tasks, offering little insight into the role of self-attention in practice.

**Network architecture design via unrolled optimization.** It is commonly believed that the success of modern deep networks largely stems from their ability to transform the raw data into compact and structured representations, which facilitates downstream tasks (Chan et al., 2022; Chen et al., 2023; Ma et al., 2022; Yu et al., 2024). A principled and interpretable approach to learning such representations with transformers is to construct an architecture that incrementally transforms tokens into these representations via unrolling optimization steps as layers of a deep network (Chan et al., 2022; Monga et al., 2021; Wang et al., 2016; Yu et al., 2023; Zhang & Ghanem, 2018). Notably, Monga et al. (2021) demonstrate that such unrolled networks are more interpretable, parameter-efficient, and effective compared to generic networks. Using this approach, each iteration of an algorithm for learning compact and structured representations is represented as one layer of deep networks. For example, Gregor & LeCun (2010) have demonstrated that sparse coding algorithms, such as ISTA, can be used to construct MLPs. Recently, Chan et al. (2022) constructed a "white-box" network based on an iterative gradient descent scheme to optimize the maximal coding rate reduction objective. More recently, Yu et al. (2024) designed a "white-box" transformer architecture by implementing an approximate alternating minimization to optimize the sparse rate reduction objective. The proposed transformer achieves performance comparable to some popular ones such as ViT (Dosovitskiy et al., 2021) and DINO (Caron et al., 2021).

## B. Proof of Theorem 3.1

### B.1. Preliminary Results

To prove Theorem 3.1, we first establish several probabilistic results about Gaussian random vectors. First, we present a probabilistic bound on the deviation of the norm of Gaussian random vectors from its mean. This is an extension of Vershynin (2018, Theorem 3.1.1).

**Lemma B.1.** *Let* $\boldsymbol{x} \sim \mathcal{N}(\boldsymbol{0}, \delta^2 \boldsymbol{I}_d)$ *be a Gaussian random vector. It holds with probability at least* $1 - 2\exp\left(-t^2/2\delta^2\right)$ *that*

$$\left| \|\boldsymbol{x}\| - \delta\sqrt{d} \right| \le t + 2\delta. \tag{10}$$

Based on the above lemma, we can respectively estimate the norm of coefficients in the signal and noise parts, the products between different pairs of Gaussian random vectors, and the bounds on the soft-max values of these products.

**Lemma B.2.** *Consider the setting in Definition 2.1 with* $p = p_1 = \cdots = p_K$ *and* $N_1 = \cdots = N_K = N/K$. *Suppose that* $p \ge 16(\sqrt{\log N} + 1)^2$ *and*

$$N \ge 8\pi K^2 \log^3 N, \ \delta \le \frac{1}{8}\sqrt{\frac{\log N}{p}}. \tag{11}$$

*The following statements hold:*

*(i) With probability at least $1 - 2KN^{-1}$, we have*

$$\left| \|\boldsymbol{a}_i\| - \sqrt{p} \right| \le 2\left(\sqrt{\log N} + 1\right), \forall i \in [N], \tag{12}$$

$$\left| \|\boldsymbol{e}_{i,l}\| - \delta\sqrt{p} \right| \le 2\delta\left(\sqrt{\log N} + 1\right), \forall i \in C_k, l \ne k \in [K]. \tag{13}$$

*(ii) With probability at least $1 - 4KN^{-2}$, we have*

$$|\langle \boldsymbol{a}_i, \boldsymbol{a}_j \rangle| \le 3\sqrt{\log N}\|\boldsymbol{a}_i\|, \forall i \ne j \in C_k, k \in [K], \tag{14}$$

$$|\langle \boldsymbol{a}_i, \boldsymbol{e}_{j,l} \rangle| \le 3\sqrt{\log N}\|\boldsymbol{e}_{j,l}\|, \forall i \in C_k, j \in C_l, k \ne l \in [K], \tag{15}$$

$$|\langle \boldsymbol{e}_{i,k}, \boldsymbol{e}_{j,k} \rangle| \le 3\delta\sqrt{\log N}\|\boldsymbol{e}_{j,k}\|, \forall i \in C_l, j \in C_m, l, m \ne k. \tag{16}$$

*(iii) With probability at least $1 - 2N^{-1}$, we have*

$$\max_{i \in C_k}\langle \boldsymbol{a}_i, \boldsymbol{e}_{j,k} \rangle \ge \sqrt{\log N}\|\boldsymbol{e}_{j,k}\|, \forall j \in C_l, l \ne k \in [K]. \tag{17}$$

*(iv) With probability at least $1 - 4KN^{-1}$, we have*

$$\frac{\exp\left(\langle \boldsymbol{a}_i, \boldsymbol{e}_{j,k} \rangle\right)}{\sum_{i' \in C_k} \exp\left(\langle \boldsymbol{a}_{i'}, \boldsymbol{e}_{j,k} \rangle\right)} \le \frac{1}{2}, \forall i \in C_k, j \in C_l, k \ne l \in [K], \tag{18}$$

$$\frac{\exp\left(\langle \boldsymbol{e}_{i,k}, \boldsymbol{e}_{j,k} \rangle\right)}{\sum_{i' \ne j, i' \in C_l} \exp\left(\langle \boldsymbol{e}_{i',k}, \boldsymbol{e}_{j,k} \rangle\right)} \le \frac{1}{2}, \forall i \ne j, i \in C_l, j \in C_m, l, m \ne k. \tag{19}$$

*Proof.* (i) Applying Lemma B.1 to $\boldsymbol{a}_i \sim \mathcal{N}(\boldsymbol{0}, \boldsymbol{I}_p)$ with $t = 2\sqrt{\log N}$ yields

$$\mathbb{P}\left(\left| \|\boldsymbol{a}_i\| - \sqrt{p} \right| \le 2(\sqrt{\log N} + 1)\right) \ge 1 - 2N^{-2}.$$

This, together with the union bound, yields that (12) holds for all $i \in [N]$ with probability at least $1 - 2N^{-1}$. Using the same argument, we obtain that (13) holds for all $i \in C_k$ and $l \ne k \in [K]$ with probability at least $1 - 2(K-1)N^{-1}$. Finally, applying the union bound yields that the probability is $1 - 2KN^{-1}$.

(ii) For each pair $(i, j)$ with $i \ne j \in C_k$ and $k \in [K]$, conditioned on $\boldsymbol{a}_i$, we have $\langle \boldsymbol{a}_i, \boldsymbol{a}_j \rangle \sim \mathcal{N}(0, \|\boldsymbol{a}_i\|^2)$. According to the tail bound the Gaussian random variable, we have

$$\mathbb{P}\left(|\langle \boldsymbol{a}_i, \boldsymbol{a}_j \rangle| \ge 3\|\boldsymbol{a}_i\|\sqrt{\log N}\,\Big|\,\boldsymbol{a}_i\right) \le 2N^{-4}.$$

This, together with the union bound, implies that conditioned on $\boldsymbol{a}_i$, it holds with probability at least $1 - 2N^{-2}$ that $|\langle \boldsymbol{a}_i, \boldsymbol{a}_j \rangle| \le 2\|\boldsymbol{a}_i\|\sqrt{\log N}$ for all $i \ne j \in C_k$ and $k \in [K]$. Using the same argument, we obtain (15) and (16). Finally, applying the union bound yields the probability.

(iii) Conditioned on $\boldsymbol{e}_{j,k}$, we obtain that $X_i := \langle \boldsymbol{a}_i, \boldsymbol{e}_{j,k} \rangle / \|\boldsymbol{e}_{j,k}\| \sim \mathcal{N}(0, 1)$ for each $i \in C_k$ are i.i.d. standard normal random variables. Then, we have

$$\mathbb{P}\left(\max_{i \in C_k} X_i \ge \sqrt{\log N}\right) = 1 - \left(\mathbb{P}\left(X_1 < \sqrt{\log N}\right)\right)^{N_k}. \tag{20}$$

Using the property of the standard Gaussian random variable, we have

$$\mathbb{P}\left(X_1 \ge t\right) \ge \left(\frac{1}{t} - \frac{1}{t^3}\right)\frac{1}{\sqrt{2\pi}}\exp\left(-\frac{t^2}{2}\right).$$

Taking $t = \sqrt{\log N}$, we obtain

$$\mathbb{P}\left(X_1 \ge \sqrt{\log N}\right) = \frac{1}{\sqrt{\log N}}\left(1 - \frac{1}{\log N}\right)\frac{1}{\sqrt{2\pi}}\exp\left(-\frac{\log N}{2}\right) \ge \frac{1}{2\sqrt{2\pi N \log N}}, \tag{21}$$

where the inequality follows from $N \geq \exp(2)$. Substituting this into (20) yields

$$\mathbb{P}\left(\max_{i \in C_k} X_i \geq \sqrt{\log N}\right) \geq 1 - \left(1 - \frac{1}{2\sqrt{2\pi N \log N}}\right)^{N/K}$$

$$\geq 1 - \exp\left(-\frac{\sqrt{N}}{2K\sqrt{2\pi \log N}}\right) \geq 1 - N^{-1},$$

where the second inequality uses $1 - x \leq \exp(-x)$ for all $x > 0$ and the last inequality follows from $N \geq 8\pi K^2 \log^3 N$. This, together with the definition of $X_i$, implies (17).

(iv) Conditioned on $e_{j,k}$, we have $X_i := \langle a_i, e_{j,k} \rangle \sim \mathcal{N}(0, \|e_{j,k}\|^2)$ for each $i \in C_k$ are i.i.d. normal random variables. Suppose that (13) holds for all $i \in C_k, l \neq k \in [K]$, which happens with probability at least $1 - 2(K-1)N^{-1}$ according to (i). This implies for all $j \in C_k$ and $k \in [K]$,

$$\|e_{j,k}\| \leq \delta\left(\sqrt{p} + 2\sqrt{\log N} + 2\right) \leq \frac{3}{2}\delta\sqrt{p}, \tag{22}$$

where the last inequality follows from $p \geq 16(\sqrt{\log N} + 1)^2$ due to (11). For ease of exposition, let

$$\sigma := \|e_{j,k}\|, \ S := \sum_{i \in C_k} \exp(X_i). \tag{23}$$

Obviously, showing (18) is equivalent to proving

$$2\exp(X_i) \leq \sum_{i' \in C_k} \exp(X_{i'}) = S, \ \forall i \in C_k. \tag{24}$$

Note that $X_i/\sigma \sim \mathcal{N}(0,1)$ for all $i \in C_k$. Using the tail bound of the standard normal random variable, we have

$$\mathbb{P}\left(\frac{|X_i|}{\sigma} \geq 2\sqrt{\log N}\right) \leq 2N^{-2}, \ \forall i \in C_k.$$

This, together with the union bound, yields that it holds with probability $1 - 2N^{-1}$ that $|X_i| \leq 2\sigma\sqrt{\log N}$ for all $i \in [N]$. Using this, (22), (23), and the union bound, we obtain with probability at least $1 - 2KN^{-1}$,

$$|X_i| \leq 3\delta\sqrt{p\log N}, \ \forall i \in [N].$$

Therefore, we have

$$\exp\left(-3\delta\sqrt{p\log N}\right) \leq \exp(X_i) \leq \exp\left(3\delta\sqrt{p\log N}\right), \ \forall i \in [N]. \tag{25}$$

Using this and (23), we have

$$S \geq \frac{N}{K}\exp\left(-3\delta\sqrt{p\log N}\right).$$

This, together with (25), implies that proving (24) is sufficient to proving

$$\log N \geq 6\delta\sqrt{p\log N} + \log(2K),$$

which holds when $N \geq \max\left\{16K^4, \exp\left(64\delta^2 p\right)\right\}$ due to (11). According to the union bound, (18) holds with probability at least $1 - 2KN^{-1}$. Using the same argument, (19) holds with probability at least $1 - 2KN^{-1}$. $\square$

### B.2. Proof of Theorem 3.1

To simplify our development, let

$$\boldsymbol{M}_1 := \begin{bmatrix} \theta^2 \boldsymbol{A}_1^T \boldsymbol{A}_1 & \theta \boldsymbol{A}_1^T \boldsymbol{E}_{2,1} & \dots & \theta \boldsymbol{A}_1^T \boldsymbol{E}_{K,1} \\ \theta \boldsymbol{E}_{2,1}^T \boldsymbol{A}_1 & \boldsymbol{E}_{2,1}^T \boldsymbol{E}_{2,1} & \dots & \boldsymbol{E}_{2,1}^T \boldsymbol{E}_{K,1} \\ \vdots & \vdots & \ddots & \vdots \\ \theta \boldsymbol{E}_{K,1}^T \boldsymbol{A}_1 & \boldsymbol{E}_{K,1}^T \boldsymbol{E}_{2,1} & \dots & \boldsymbol{E}_{K,1}^T \boldsymbol{E}_{K,1} \end{bmatrix} \in \mathbb{R}^{N \times N}, \tag{26}$$

$$\boldsymbol{M}_2 := \begin{bmatrix} \boldsymbol{E}_{1,2}^T \boldsymbol{E}_{1,2} & \theta \boldsymbol{E}_{1,2}^T \boldsymbol{A}_2 & \dots & \boldsymbol{E}_{1,2}^T \boldsymbol{E}_{K,2} \\ \theta \boldsymbol{A}_2^T \boldsymbol{E}_{1,2}^T & \theta^2 \boldsymbol{A}_2^T \boldsymbol{A}_2 & \dots & \theta \boldsymbol{A}_2^T \boldsymbol{E}_{K,2} \\ \vdots & \vdots & \ddots & \vdots \\ \boldsymbol{E}_{K,2}^T \boldsymbol{E}_{1,2} & \theta \boldsymbol{E}_{K,2}^T \boldsymbol{A}_2 & \dots & \boldsymbol{E}_{K,2}^T \boldsymbol{E}_{K,2} \end{bmatrix} \in \mathbb{R}^{N \times N},$$

$$\vdots$$

$$\boldsymbol{M}_K := \begin{bmatrix} \boldsymbol{E}_{1,K}^T \boldsymbol{E}_{1,K} & \boldsymbol{E}_{1,K}^T \boldsymbol{E}_{2,K} & \dots & \theta \boldsymbol{E}_{1,K}^T \boldsymbol{A}_K \\ \boldsymbol{E}_{2,K}^T \boldsymbol{E}_{1,K} & \boldsymbol{E}_{2,K}^T \boldsymbol{E}_{2,K} & \dots & \theta \boldsymbol{E}_{2,K}^T \boldsymbol{A}_k \\ \vdots & \vdots & \ddots & \vdots \\ \theta \boldsymbol{A}_K^T \boldsymbol{E}_{1,K} & \theta \boldsymbol{A}_K^T \boldsymbol{E}_{2,K} & \dots & \theta^2 \boldsymbol{A}_K^T \boldsymbol{A}_K \end{bmatrix} \in \mathbb{R}^{N \times N}.$$

where $\theta \geq 1$. Recall that

$$\boldsymbol{Z}^{(0)} = \begin{bmatrix} \boldsymbol{Z}_1^{(0)} & \dots & \boldsymbol{Z}_K^{(0)} \end{bmatrix} = \begin{bmatrix} \boldsymbol{U}_1 \boldsymbol{A}_1 + \sum_{j \neq 1} \boldsymbol{U}_j \boldsymbol{E}_{1,j} & \dots & \boldsymbol{U}_K \boldsymbol{A}_K + \sum_{j \neq K} \boldsymbol{U}_j \boldsymbol{E}_{K,j} \end{bmatrix}, \tag{27}$$

**Lemma B.3.** *Consider the setting in Definition 2.1 with $p = p_1 = \dots = p_K$ and $N_1 = \dots = N_K = N/K$. Let $\varphi(\cdot)$ be*

$$\varphi(\boldsymbol{x}) = h(\sigma(\boldsymbol{x})), \tag{28}$$

*where $\sigma : \mathbb{R}^N \to \mathbb{R}^N$ is the soft-max function and $h : \mathbb{R}^N \to \mathbb{R}^N$ is an element-wise thresholding function with $h(x) = \tau \mathbb{I} \{x > \tau\}$ for each $i \in [N]$. Suppose that (11) holds. Suppose in addition that $p \geq 64(\sqrt{\log N} + 1)^2$ and*

$$\tau \in \left( \frac{1}{2}, \frac{1}{1 + N \exp(-9p/32)} \right] \tag{29}$$

*The following statements hold with probability at least $1 - KN^{-\Omega(1)}$ that ,*

$$\varphi(\boldsymbol{M}_1) = \mathrm{BlkDiag}(\tau \boldsymbol{I}, \boldsymbol{0}, \dots, \boldsymbol{0}), \ \dots, \ \varphi(\boldsymbol{M}_K) = \mathrm{BlkDiag}(\boldsymbol{0}, \boldsymbol{0}, \dots, \tau \boldsymbol{I}). \tag{30}$$

*Proof.* Suppose that (12)-(19) hold, which happens with probability at least $1 - KN^{-\Omega(1)}$ according to Lemma B.2, (11), and the union bound. Now, we focus on studying $\boldsymbol{M}_1$ as defined in (26). For ease of exposition, we denote the $i$-th column of $\boldsymbol{M}_1$ by $\boldsymbol{m}_i \in \mathbb{R}^N$ for each $i \in [N]$. Moreover, recall that

$$C_1 = \left\{ 1, 2, \dots, \frac{N}{K} \right\}, \dots, C_K = \left\{ \frac{(K-1)N}{K} + 1, \frac{(K-1)N}{K} + 2, \dots, N \right\}.$$

We now divide our proof into two cases. We first study the $i$-th column of $\boldsymbol{M}_1$ for each $i \in C_1$, and then study the $i$-th column of $\boldsymbol{M}_1$ for each $i \in C_k$ with $k \neq 1$.

**Case 1.** According to (26), we have for each $i \in C_1$,

$$m_{ij} = \theta^2 \langle \boldsymbol{a}_i, \boldsymbol{a}_j \rangle, \forall j \in C_1, \ m_{ij} = \theta \langle \boldsymbol{a}_i, \boldsymbol{e}_{j,k} \rangle, \forall j \in C_k, k \neq 1.$$

For each pair $(i, j)$ with $i \neq j \in C_1$, we compute

$$\frac{\sigma_i(\boldsymbol{m}_i)}{\sigma_j(\boldsymbol{m}_i)} = \exp\left( m_{ii} - m_{ij} \right) \geq \exp\left( \theta \|\boldsymbol{a}_i\| \left( \theta \|\boldsymbol{a}_i\| - 3\sqrt{\log N} \right) \right) \geq \exp\left( \frac{9\theta^2 p}{32} \right), \tag{31}$$

where the first inequality follows from (14) and the second uses (12) and $\sqrt{p} \geq 8(\sqrt{\log N} + 1)$. Using the same argument, for each pair $(i,j)$ with $i \in C_1$, $j \in C_k$, and $k \neq 1$, we obtain

$$\frac{\sigma_i(\boldsymbol{m}_i)}{\sigma_j(\boldsymbol{m}_i)} \geq \exp\left(\frac{9\theta^2 p}{32}\right),$$

This, together with $\sum_{j=1}^{N} \sigma_j(\boldsymbol{m}_i) = 1$, yields $\left(1 + (N-1)\exp\left(-9\theta^2 p/32\right)\right)\sigma_i(\boldsymbol{m}_i) \geq 1$. Therefore, we have for each $i \in C_1$,

$$\sigma_i(\boldsymbol{m}_i) \geq \frac{1}{1 + N\exp(-9\theta^2 p/32)} > \frac{1}{2}, \ \sigma_j(\boldsymbol{m}_i) \leq \frac{1}{2}, \ \forall j \neq i, \tag{32}$$

where the last inequality follows from $p \geq 64(\sqrt{\log N} + 1)^2$. This, together with the value of $\tau$ in (29), yields for each $i \in C_1$,

$$\sigma_j(\boldsymbol{m}_i) < \tau < \sigma_i(\boldsymbol{m}_i), \ \forall j \neq i.$$

Using this and (28), we have for each $i \in C_1$,

$$h\left(\sigma_i(\boldsymbol{m}_i)\right) = \tau, \ h\left(\sigma_j(\boldsymbol{m}_i)\right) = 0, \ \forall j \neq i.$$

**Case 2.** For each $i \in C_k$ with $k \neq 1$, it follows from (26) that

$$m_{ij} = \theta\langle \boldsymbol{e}_{i,1}, \boldsymbol{a}_j \rangle, \forall j \in C_1, \ m_{ij} = \langle \boldsymbol{e}_{i,1}, \boldsymbol{e}_{j,1} \rangle, \ \forall j \in C_l, l \neq 1.$$

Consider a fixed $i \in C_k$ with $k \neq 1$, it follows from (17) that there exists $j_i \in C_1$ such that $m_{ij_i} \geq \theta\|\boldsymbol{e}_{i,1}\|\sqrt{\log N}$. This implies

$$\frac{\sigma_{j_i}(\boldsymbol{m}_i)}{\sigma_i(\boldsymbol{m}_i)} = \exp\left(\theta m_{ij_i} - m_{ii}\right) \geq \exp\left(\|\boldsymbol{e}_{i,1}\|\left(\theta\sqrt{\log N} - \|\boldsymbol{e}_{i,1}\|\right)\right)$$

$$\geq \exp\left(\frac{3\delta\theta}{4}\sqrt{p\log N} - \frac{25}{16}\delta^2 p\right),$$

where the second inequality follows from (13). This, together with $\sigma_i(\boldsymbol{m}_i) + \sigma_{j_i}(\boldsymbol{m}_i) < 1$, implies

$$\sigma_i(\boldsymbol{m}_i) < \frac{1}{1 + \exp\left(3\delta\theta\sqrt{p\log N}/4 - 25\delta^2 p/16\right)} < \frac{1}{1 + \exp\left(\delta\theta\sqrt{p\log N}/2\right)} < \frac{1}{2}, \tag{33}$$

where the second inequality uses $\delta\sqrt{p} \leq \sqrt{\log N}/8$ due to (11). On the other hand, it follows from (18) and (19) that

$$\sigma_j(\boldsymbol{m}_i) \leq \frac{1}{2}, \forall j \neq i.$$

This, together with (33), $\delta \leq 1/8$, $\sqrt{p} \geq 8(\sqrt{\log N} + 1)$, and the value of $\tau$ by (29), yields for each $i \in C_k$ with $k \neq 1$,

$$\sigma_j(\boldsymbol{m}_i) < \tau, \forall j \in [N]. \tag{34}$$

This directly implies

$$h\left(\sigma(\boldsymbol{m}_i)\right) = \boldsymbol{0}, \ \forall i \in C_k, k \neq 1.$$

Then, we have $\varphi(\boldsymbol{M}_1) = \begin{bmatrix} \tau \boldsymbol{I} & \boldsymbol{0} \\ \boldsymbol{0} & \boldsymbol{0} \end{bmatrix}$. Applying the same argument to $\boldsymbol{M}_2, \ldots, \boldsymbol{M}_K$, we obtain (30). $\qquad\square$

Armed with the above result, we are ready to prove Theorem 3.1.

*Proof of Theorem 3.1.* For ease of exposition, let $\boldsymbol{M}_k^{(l)} := \boldsymbol{Z}^{(l)^T} \boldsymbol{U}_k \boldsymbol{U}_k^T \boldsymbol{Z}^{(l)}$ for each $k \in [K]$ and $l \in [L]$. Suppose that (30) holds, which happens with probability at least $1 - K N^{-\Omega(1)}$ according to (11), and (29), Lemma B.3. We claim that for each $l \in [L]$, we have

$$\boldsymbol{Z}^{(l)} = \left[ (1 + \eta\tau)^l \boldsymbol{U}_1 \boldsymbol{A}_1 + \sum_{j \neq 1} \boldsymbol{U}_j \boldsymbol{E}_{1,j} \quad \ldots \quad (1 + \eta\tau)^l \boldsymbol{U}_K \boldsymbol{A}_K + \sum_{j \neq K} \boldsymbol{U}_j \boldsymbol{E}_{K,j} \right]. \tag{35}$$

This, together with (6), yields for each $k \in [K]$ and $l \in [L]$,

$$\mathrm{SNR}(\boldsymbol{Z}_k^{(l)}) = \frac{\|\boldsymbol{U}_k \boldsymbol{U}_k^T \boldsymbol{Z}_k^{(l)}\|_F}{\|(\boldsymbol{I} - \boldsymbol{U}_k \boldsymbol{U}_k^T) \boldsymbol{Z}_k^{(l)}\|_F} = \frac{(1 + \eta\tau)^l \|\boldsymbol{A}_k\|_F}{\|\sum_{j \neq k} \boldsymbol{U}_j \boldsymbol{E}_{k,j}\|_F},$$

which directly implies (8) for each $k \in [K]$ and $l \in [L-1]$. According to the union bound, the probability is $1 - KLN^{-\Omega(1)}$.

The rest of the proof is devoted to proving the claim (35) using the induction method. First, we consider the base case $l = 1$. According to (27) and (7), we compute

$$\boldsymbol{U}_1 \boldsymbol{U}_1^T \boldsymbol{Z}^{(0)} = \begin{bmatrix} \boldsymbol{U}_1 \boldsymbol{A}_1 & \boldsymbol{U}_1 \boldsymbol{E}_{2,1} & \ldots & \boldsymbol{U}_1 \boldsymbol{E}_{K,1} \end{bmatrix},$$

$$\boldsymbol{M}_1^{(0)} = (\boldsymbol{U}_1 \boldsymbol{U}_1^T \boldsymbol{Z}^{(0)})^T (\boldsymbol{U}_1 \boldsymbol{U}_1^T \boldsymbol{Z}^{(0)}) = \begin{bmatrix} \boldsymbol{A}_1^T \boldsymbol{A}_1 & \boldsymbol{A}_1^T \boldsymbol{E}_{2,1} & \ldots & \boldsymbol{A}_1^T \boldsymbol{E}_{K,1} \\ \boldsymbol{E}_{2,1}^T \boldsymbol{A}_1 & \boldsymbol{E}_{2,1}^T \boldsymbol{E}_{2,1} & \ldots & \boldsymbol{E}_{2,1}^T \boldsymbol{E}_{K,1} \\ \vdots & \vdots & \ddots & \vdots \\ \boldsymbol{E}_{K,1}^T \boldsymbol{A}_1 & \boldsymbol{E}_{K,1}^T \boldsymbol{E}_{2,1} & \ldots & \boldsymbol{E}_{K,1}^T \boldsymbol{E}_{K,1} \end{bmatrix}.$$

Using the same argument, we can compute $\boldsymbol{M}_k^{(0)}$ for each $k \in [K]$. This, together with (30) for each $k \in [K]$, yields

$$\sum_{k=1}^K \boldsymbol{U}_k \boldsymbol{U}_k^T \boldsymbol{Z}^{(0)} \varphi(\boldsymbol{M}_k^{(0)}) = \begin{bmatrix} \tau \boldsymbol{U}_1 \boldsymbol{A}_1 & \tau \boldsymbol{U}_2 \boldsymbol{A}_2 & \ldots & \tau \boldsymbol{U}_K \boldsymbol{A}_K \end{bmatrix}.$$

Using this, (27), and (4), we directly obtain that (35) holds for $l = 1$. Next, we consider the case $l \geq 2$. Suppose that (35) holds for some $l \geq 1$. We compute

$$\boldsymbol{U}_1 \boldsymbol{U}_1^T \boldsymbol{Z}^{(l)} = \begin{bmatrix} (1 + \eta\tau)^l \boldsymbol{U}_1 \boldsymbol{A}_1 & \boldsymbol{U}_1 \boldsymbol{E}_{2,1} & \ldots & \boldsymbol{U}_1 \boldsymbol{E}_{K,1} \end{bmatrix},$$

$$\boldsymbol{M}_1^{(l)} = \begin{bmatrix} (1 + \eta\tau)^{2l} \boldsymbol{A}_1^T \boldsymbol{A}_1 & (1 + \eta\tau)^l \boldsymbol{A}_1^T \boldsymbol{E}_{2,1} & \ldots & (1 + \eta\tau)^l \boldsymbol{A}_1^T \boldsymbol{E}_{K,1} \\ (1 + \eta\tau)^l \boldsymbol{E}_{2,1}^T \boldsymbol{A}_1 & \boldsymbol{E}_{2,1}^T \boldsymbol{E}_{2,1} & \ldots & \boldsymbol{E}_{2,1}^T \boldsymbol{E}_{K,1} \\ \vdots & \vdots & \ddots & \vdots \\ (1 + \eta\tau)^l \boldsymbol{E}_{K,1}^T \boldsymbol{A}_1 & \boldsymbol{E}_{K,1}^T \boldsymbol{E}_{2,1} & \ldots & \boldsymbol{E}_{K,1}^T \boldsymbol{E}_{K,1} \end{bmatrix}.$$

Using the same argument, we can compute $\boldsymbol{M}_k^{(l)}$ for each $k \in [K]$. This, together with (30) for each $k \in [K]$, yields

$$\sum_{k=1}^K \boldsymbol{U}_k \boldsymbol{U}_k^T \boldsymbol{Z}^{(0)} \varphi(\boldsymbol{M}_k^{(0)}) = \begin{bmatrix} (1 + \eta\tau)^l \tau \boldsymbol{U}_1 \boldsymbol{A}_1 & (1 + \eta\tau)^l \tau \boldsymbol{U}_2 \boldsymbol{A}_2 & \ldots & (1 + \eta\tau)^l \tau \boldsymbol{U}_K \boldsymbol{A}_K \end{bmatrix}.$$

Using this, (27), and (4), we directly obtain that (35) holds for $l + 1$. Then, we prove the claim. $\qquad \square$

*Table 4.* The architectures of GPT-2 and AoT models. For GPT-2, each layer consists of an attention operator and an MLP, while each layer only has an attention operator in AoT.

| Models | Num. of para. | Num. of layers | Embedding dim. | Num. of heads | Attention type | Has MLP |
|---|---|---|---|---|---|---|
| AoT-MSSA-L Base | 102M | 24 | 1024 | 16 | MSSA | No |
| AoT-MSSA-L Medium | 182M | 36 | 1280 | 20 | MSSA | No |
| AoT-MHSA-L Base | 122M | 24 | 896 | 14 | MHSA | No |
| GPT-2 Base | 124M | 12 | 768 | 12 | MHSA | Yes |

# C. Additional Experiemental Details

## C.1. Language Model Configuration

We provide details of the language model architecture in Table 4

## C.2. ICL Configuration

We study decoder-only transformer models in GPT-2 family (Radford et al., 2019) and its corresponding AoT variants. As in Park et al. (2024a), we perform the same grid search over learning rates in $\{10^{-4}, 5 \times 10^{-5}, 2 \times 10^{-4}, 4 \times 10^{-4}\}$, and clipping the gradient norm in $\{5.0, 10.0, 50.0\}$.

*Table 5.* The detailed architectures of transformer and AoT used in ICL experiments. To ensure a fair comparison, all AoT models are designed with a larger number of layers to match the size of the transformer.

| Models | Num. of para. | Num. of layers | Embedding dim. | Num. of heads | Attention type | Has MLP |
|---|---|---|---|---|---|---|
| AoT-MSSA-L | 7.5M | 32 | 128 | 8 | MSSA | No |
| AoT-MHSA-L | 8.55M | 32 | 128 | 8 | MHSA | No |
| Transformer | 9.63M | 16 | 128 | 8 | MHSA | Yes |

## C.3. Emergence of Semantic Meaning

The attention heads in our models have different semantic meanings, and indeed demonstrate the interpretability of our proposed architecture in practice. In Figure 7, we train the AoT model with the MSSA operator on ImageNet-1K and visualize the self-attention heatmaps between the [CLS] token and other image patches. Note that the [CLS] token is the "class token", a trainable model parameter inserted along with other image tokens to represent the class information. We select 5 attention heads by manual inspection and find that they capture different parts of objects, displaying different semantic meanings.

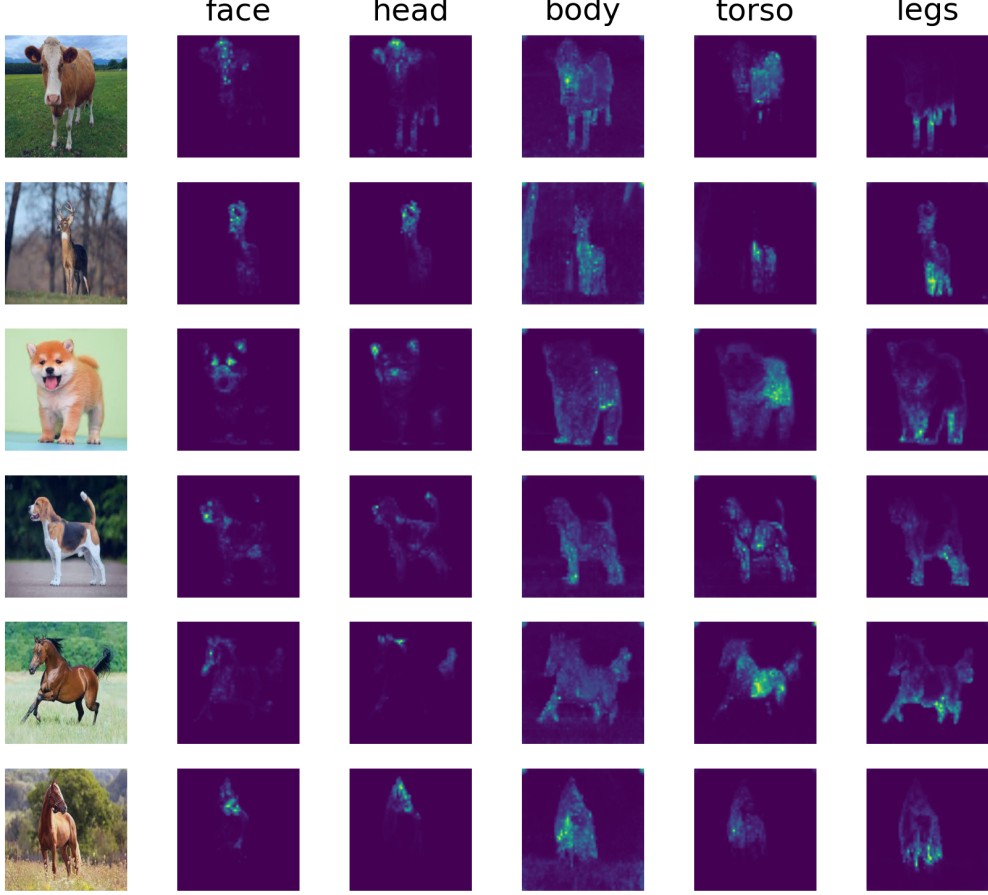

*Figure 7.* **Visualization of attention heads on ImageNet-1K.** We feed a trained AoT-MSSA a mini-batch of images and extract the attention maps of different heads from the penultimate layer. We show that these heads capture certain semantic meanings across different images.

