# OpenReview forum: "Attention-Only Transformers via Unrolled Subspace Denoising"
_ICML.cc/2025/Conference — ICML 2025 poster_

### Official Review · Reviewer_sksF · 2025-03-11

**Overall Recommendation:** 4

**Summary:**

This paper presents a new transformer architecture by interpreting each layer as a subspace denoising operator for the token representations. The work is interesting, and the paper is well written in general. The authors also provide theoretical analysis on the developed model. Experimental results show that the proposed method can yield comparable performance in different applications.

**Claims And Evidence:**

The experiments support the claims in the paper.

**Essential References Not Discussed:**

N/A

**Experimental Designs Or Analyses:**

The experiments are sufficient to support the claims in the paper. However, it could further benefit from comparison with more recent SOTA transformer-based methods.

**Methods And Evaluation Criteria:**

The proposed methods and evalutions make sense in the paper.

**Other Comments Or Suggestions:**

N/A

**Other Strengths And Weaknesses:**

Strengths: 1) Very good presentation of the work and the motivation is clear. 2) interesting findings to interpret a layer of transformer as a subspace denoising operator for token representation. 3) the work is theoretically founded and empirically verified in different tasks.

Weaknesses: 1) More fair discussions should be given in the experiments. For instance, the results in Table 1 show that the proposed method is 9.3% lower than the compared baseline. Using "comparable performance" is unfair.
2) The advantage of the proposed method might be strengthened by comparing with more transformer-based methods.

**Questions For Authors:**

See the weaknesses above.

**Relation To Broader Scientific Literature:**

The idea of interpreting each layer of transformer as a subspace denoising operator for token representations is interesting, which facilitates the understanding of the commonly used transformer network.

**Theoretical Claims:**

The authors provide proofs to some of the theories.

---

> ### Author Rebuttal · Authors · 2025-04-01
>
> > **Q1.** More fair discussions should be given in the experiments. For instance, the results in Table 1 show that the proposed method is 9.3% lower than the compared baseline. Using "comparable performance" is unfair.
>
> **A1.** Thanks for the comment. To address the concern, we will change "comparable performance" to ''worse performance'' and revise the manuscript to provide a more accurate description of the results.
>
> Additionally, after a more careful examination, we found that the CRATE baseline we compared against uses a patch size of 8, while our implementation uses a patch size of 16, making the comparison unfair. A smaller patch size results in more image tokens, which usually leads to better performance.  We are currently training models with a patch size of 8 and will report the result shortly.
>
> > **Q2.** The advantage of the proposed method might be strengthened by comparing it with more transformer-based methods.
>
> **A2.** Thanks for the suggestion. As suggested, we first compare AoT-MHSA to Vision Transformer (ViT) [1] on the ImageNet. Due to the time limitation,  we train the model on ImageNet-1K from scratch and report the results as follows:
>
> | Model | # of Parameters | Accuracy |
> | -------- | -------- | -------- |
> | AoT-MHSA     |  15M    | 69.5%     |
> | ViT Small     | 22M     | 72.4%     |
>
> Next, we compare AoT-MSSA to Llama [2] architecture on the in-context learning task (Section 4.2.2) and report the result at the following anonymous link: https://postimg.cc/gallery/d0Ypwvc.
>
> Based on the above experimental results, it is observed that the proposed architecture demonstrates performance that is comparable to, or slightly lower than, that of state-of-the-art transformers.
>
> *[1] Dosovitskiy, Alexey, et al. An Image is Worth 16x16 Words: Transformers for Image Recognition at Scale. In International Conference on Learning Representations, 2021.*
>
> *[2] Touvron, Hugo, et al. Llama: Open and efficient foundation language models. arXiv preprint arXiv:2302.13971.*

---

### Official Review · Reviewer_5Af5 · 2025-03-14

**Overall Recommendation:** 3

**Summary:**

The authors propose an attention-only architecture, using the multi-head subspace self-attention (MSSA), first proposed by Yu et al., NeurIPS 2023 ( and also JMLR 2024). They have a model in which the embeddings of the tokens come from a mixture of different subspaces, albeit with additional additive noise making each token slightly away from its natural subspace. The purpose of the network is to denoise these vectors at each layer, increasing the signal to noise ratio as one goes from layer to layer. They view their architecture as a special case of a multi-head attention model.

The authors provide some theoretical analysis assuming a particular model for $Z^{(l)}$. They try this architecture for vision transformer for ImageNet top-1 accuracy and then also for language models doing next token prediction. The performances are not meant to be competitive with the best engineered transformer models, but are still nontrivial. They also do some in-context learning for linear models. The number of parameters are only in tens of millions as opposed to billions.

## update after rebuttal

I am satisfied with the authors' clarifications. I am keeping the same score.

**Claims And Evidence:**

I take that the authors' main claim is that transformers can act as denoisers which ultimately provide a compressed representation allowing easy training for downstream tasks. I think they have shown some evidence for this claim. They also claim the architecture is interpretable, and show that different heads have some semantic interpretability (Fig. 7, Appendix D).

**Essential References Not Discussed:**

I am not aware of such references.

**Experimental Designs Or Analyses:**

Experimental design and analysis seems sufficient.

**Methods And Evaluation Criteria:**

Once more, the benchmarks and the datasets are less demanding that state-of-the-art transformers.

**Other Comments Or Suggestions:**

None

**Other Strengths And Weaknesses:**

There are some notations in the paper that I think would confuse the first time reader. $Z^{(l)}, f^{l}$ have layer superscript $l$ in the equations but $U_k$ never does, giving the impression that $U_k$ are the same for each layer. Figure 3 architecture has $U^{l}$ and so does Yu et al, 2023. Unfortunately, Figure 2 does the conceptual explaining with $U_k$'s apparently the same for each layer. Also, layer norm appears in Fig. 3 but does not get mentioned much in the model or the analysis.

**Questions For Authors:**

I would prefer a clearer presentation of the architecture, and not having to rely on the previous papers for disambiguation.

**Relation To Broader Scientific Literature:**

How transformers form such a powerful representation of sequence is something of a mystery. The authors are advancing a view of transformers as a signal denoiser via something like a subspace clustering. This MLP layer free transformer is easier to analyze a provide some credence for this view. Of course, its performance is going to be far from satisfactory. We have to see if explainability/interpretability compensates for the loss of performance.

**Theoretical Claims:**

I have not checked the proofs in Appendix B in detail but got a sense of the techniques involved and the general flow of the argument.

---

> ### Author Rebuttal · Authors · 2025-04-01
>
> > **Q1.** **Inconsistent notation**: $Z^{(l)}$, $f^l$ have layer superscript $l$ in the equation but $U_k$ never does, giving the impression that $U_k$ are the same for each layer. Figure 3 architecture has $U^l$ and so does Yu et al., 2023.
>
> **A1.** Thanks for pointing this out. This confusion stems from a gap between the theoretical construction of transformers under Definition 2.1 and their practical implementation for real-world data. Specifically, our approach constructs a transformer in a forward manner by interpreting each layer as a subspace denoising operator, assuming that the initial token representations satisfy Definition 2.1. In this theoretical setting,  it suffices to consider $U_k^{(l)} = U_k$ across layers. However, in practical scenarios with real-world data, these subspace matrices $\{U_k^{(l)}\}$ need to be learned gradually via backpropagation and may be different across layers. We will include the above discussion and the following equation in Section 3.1 to clarify this point:
> $$
>  \mathrm{Z}^{(l+1)} =  \mathrm{Z}^{(l)} + \eta \sum_{k=1}^K  \mathrm{U}_k^{(l)} \mathrm{U}_k^{(l)^T}  \mathrm{Z}^{(l)}\varphi\left( \mathrm{Z}^{(l)^T}\mathrm{U}_k^{(l)}\mathrm{U}_k^{(l)^T}\mathrm{Z}^{(l)}\right)
> $$
>
> > **Q2.** **Layer norm:**  Layer norm appears in Fig. 3 but does not get mentioned much in the model or the analysis.
>
> **A2.** Yes, layer normalization indeed appears in Fig. 3 as part of the practical implementation of the transformer architecture. Our theoretical framework primarily focuses on the core components of transformers, namely self-attention and skip connections, as denoising token representations can be achieved without layer normalization when the initial token representations satisfy Definition 2.1. However, in practice, layer normalization is necessary to stabilize training and improve convergence. We will include a brief discussion on the role of layer norm in the revised version to clarify its inclusion in the architecture.
>
>
> > **Q3.** I would prefer a clearer presentation of the architecture and not having to rely on the previous papers for disambiguation.
>
> **A3.** Thanks for the valuable comment. To address this, we will include a more detailed and explicit description of the architecture in the revised manuscript to reduce the dependency on previous papers such as Yu et al. (2023a, b).

---

> > ### Comment · Reviewer_5Af5 · 2025-04-04
> >
> > I thank the authors for their clarificatory comments. I would like to keep the same score.

---

> > > ### Author Response · Authors · 2025-04-07
> > >
> > > We sincerely appreciate the reviewer's acknowledgment of our clarifications. We believe the improvements presented in the rebuttal further enhance the clarity of the paper!

---

### Official Review · Reviewer_KY7F · 2025-03-15

**Overall Recommendation:** 3

**Summary:**

This paper proposes an attention-only transformer (AoT) architecture that eliminates the feed-forward network (FFN) modules found in traditional transformers, including CRAFT's Multi-head Subspace Self-Attention (MSSA). The authors argue that representation learning should compress noisy token representations toward a mixture of low-dimensional subspaces, and that multi-head self-attention naturally implements this denoising operation.

**Claims And Evidence:**

## Strengths:
1. The paper is well-written. Figures are clear to understand the paper.
2. The theoretical formulation is clear, particularly regarding the concept of a union of (possibly many) low-dimensional subspaces.
3. The paper provides a solid mathematical foundation for its claims.

## Weaknesses:

1. There appears to be an inconsistency between theory and implementation. According to the parameter calculations, MSSA ultimately employs the same methodology as MHSA, where all projection matrices U_o, U_q, U_k, and U_v are not shared. This contradicts the theoretical framework presented, suggesting a mismatch between the proposed theory and actual implementation. I am not sure about this point, I do not check the authors' code.

2. The experimental results in Table 1 indicate that AoT's performance is significantly suboptimal. On ImageNet, improvements of 0.2 or more are typically considered statistically significant, with 20M-30M models usually achieving accuracy between 80-84%. However, AoT only reaches approximately 70%, making it difficult to consider this approach effective.

3. The claims regarding experimental results are problematic. In nanogpt experiments, according some previous top-conference papers, improvements of 0.02-0.03 in validation loss are typically considered meaningful. While increasing network parameters generally leads to lower validation loss, the paper shows that under comparable parameter conditions, AoT's loss is approximately 0.1 worse than the baseline, yet the authors claim comparable performance, not fair claim. Even more concerning, AoT with 180M parameters performs worse than the baseline (124M), strongly suggesting that AoT is an ineffective or significantly underperforming approach.

4. There is a dimensional inconsistency in Equation 5. Since head_k is d_h by n, head_k^T should be n by d_h, but the matrix dimensions do not align properly in the equation.

5. The mathematical foundations of the paper are difficult to fully verify as they require extensive knowledge of high-dimensional probability theory and considerable time investment. Upon brief examination of Lemma B.1, potential errors were identified. For example, when t=0, delta=1, and d=64, the lemma yields an impossible probability value. This suggests that additional constraints on t or other parameters may be necessary for the lemma to hold true.

**Essential References Not Discussed:**

references are ok.

**Ethical Review Concerns:**

no ethical concerns

**Experimental Designs Or Analyses:**

yes, it is ok, but the experimental results are not good enough.

**Methods And Evaluation Criteria:**

Yes, it makes sense

**Other Comments Or Suggestions:**

see above

**Other Strengths And Weaknesses:**

see above

**Questions For Authors:**

see above

**Relation To Broader Scientific Literature:**

It provides a new perspective of understanding self-attention.

**Theoretical Claims:**

I partly check its proofs, I mention some problems.

---

> ### Author Rebuttal · Authors · 2025-04-01
>
> > **Weakness 1.** **Inconsistency between theory and implementation**: According to the parameter calculations, MSSA ultimately employs the same methodology as MHSA, where all projection matrices $U_o$, $U_q$, $U_k$, and $U_v$ are not shared...
>
> **A1.** We should clarify that the theory and implementation are consistent. Specifically, we have implemented transformers using both MSSA (see AoT-MSSA in Section 4.1) and MHSA (see AoT-MHSA in Section 4.2). Our implementation of AoT-MSSA strictly follows the theoretical framework presented in the paper, where the projection matrices are designed according to the structured attention mechanism.
>
> > **Weakness 2.** **Suboptimal experimental results**: The experimental results in Table 1 indicate that AoT's performance is significantly suboptimal.
>
> **A2.** Thanks for the comment. After a more careful examination, we found that the CRATE baseline we compared against uses a patch size of 8 in Table 1, while our implementation uses a patch size of 16, making the comparison unfair. A smaller patch size results in more image tokens, which usually leads to better performance.  We are currently training models with a patch size of 8 and will report the result shortly.
>
> Moreover, we compare AoT-MHSA to Vision Transformer (ViT) [1] on the ImageNet. We train the model on ImageNet-1K from scratch and report the results as follows:
>
> | Model | # of Parameters | Accuracy |
> | -------- | -------- | -------- |
> | AoT-MHSA     |  15M    | 69.5%     |
> | ViT Small     | 22M     | 72.4%     |
>
> We believe that with further tuning, we can narrow the performance gap even more.
>
> *[1] Dosovitskiy, Alexey, et al. An Image is Worth 16x16 Words: Transformers for Image Recognition at Scale. In International Conference on Learning Representations, 2021.*
>
> > **Weakness 3.** **Problematic claims on the experimental results:** In nanogpt experiments, according to some previous top-conference papers, improvements of 0.02-0.03 in validation loss are typically considered meaningful...
>
> **A3.**  Thank you for the feedback. We have retrained AoT-MHSA with additional hyperparameter tuning and present the updated experimental results below:
>
> | Model (# of Parameters) | LAMBADA (val loss) | PTB  (val loss) | WikiText  (val loss) | LAMBADA (acc) | CBT CN (acc)| CBT NE (acc) | OWT (val loss) |
> | -------- | -------- | -------- | -------- | -------- | -------- | -------- | -------- |
> | AoT-MHSA (122M)    | 4.42    | 5.52 | 4.19 | 0.38 | 0.86 | 0.82 |2.92     |
> | GPT2 (124M)     | 4.32    | 5.75  | 4.13 | 0.4 | 0.87 | 0.84 |  2.86     |
>
>
> We acknowledge that our method is still underperforming. However, compared to Table 2, the performance of the proposed transformer has shown improvement than before, achieving a smaller validation loss. Here, its full potential has not been fully explored due to limited computing resources. With further hyperparameter tuning, we believe that performance can be further improved. Additionally, the primary focus of this paper is on the theoretical contributions rather than empirical performance.
>
> > **Weakness 4.** **Inconsistency in Eq. (5):** There is a dimensional inconsistency in Equation 5. Since $\mathrm{head}_k$ is d_h by n, $\mathrm{head}_k^T$ should be n by d_h, but the matrix dimensions do not align properly in the equation.
>
> **A4.** Thanks for catching this error. To fix it, we revise Eq. (5) into
> $$
> \mathrm{MHSA}(Z) = W_O \begin{bmatrix}
> \mathrm{head}_1 \\\\ \dots \\\\  \mathrm{head}_K
> \end{bmatrix}.
> $$
> Here, $Z \in \mathbb{R}^{d\times n}$, $W^O \in \mathbb{R}^{d\times Kd_h}$, and $\begin{bmatrix}
> \mathrm{head}_1 \\\\ \dots \\\\  \mathrm{head}_K
> \end{bmatrix} \in \mathbb{R}^{Kd_h \times n}$ due to each head $ \mathrm{head}_k \in \mathbb{R}^{d_h\times n}$.
>
> > **Weakness 5.** **Difficult mathematical theory**: Upon brief examination of Lemma B.1, potential errors were identified. For example, when $t=0$, $\delta=1$, and $d=64$, the lemma yields an impossible probability value. This suggests that additional constraints on t or other parameters may be necessary for the lemma to hold true.
>
> **A5.** Thanks for the comment. We will provide additional explanations and clarification to facilitate understanding in the revised manuscript. The result in Lemma B.1 is a standard concentration inequality for Gaussian random vectors; see Theorem 5.6 & Example 5.7 in Ref [2].
>
> It is important to note that the choice of $t \ge 0$ is crucial for the validity of the inequality. Setting t=0 would result in a trivial statement, as the inequality is designed to quantify significant deviations from the mean. Therefore, only when the deviation is sufficiently large does the inequality yield a meaningful result. For example, when $\delta=1$ and we set $t=2\sqrt{\log d}$, then it holds with probability at least $1-2/d^2$ that $||x-\sqrt{d}|| \le \sqrt{2\log d}+2$.
>
>
> *[2] Boucheron et al. (2013). Concentration Inequalities: A Nonasymptotic Theory of Independence. Oxford University Press.*

---

### Decision · Program_Chairs · 2025-05-01

**Decision:**

Accept (poster)

**Comment:**

This paper provides an interpretation of self attention layers as an unrolled iterative denoising process, and proposes a Transformer only architecture with a MSSA attention layer. They demonstrate that this architecture achieves non-trivial performance on a set of benchmarks, which lays a foundation for understanding Transformers in a simpler setting. Reviewers in general appreciate the angle the paper takes, and acknowledges the value of such study despite its lack of competitive performance with modern Transformers. Further clarifications and results in the rebuttal also helped resolve some of the concerns. The AC agrees with the reviewers, but also thinks that the paper can be further improved by better elaborating on the potential benefit of the denoising view of Transformers, or how that could help us better understand standard Transformers. Given these considerations, the AC leans towards accepting this work if there is space.